# Insight into the Structure of Antifungal Cyrmenins: Conformational Studies of Unique Dehydroamino Acid, O-Methyldehydroserine

**DOI:** 10.3390/ijms26010340

**Published:** 2025-01-02

**Authors:** Karolina Banaś, Paweł Lenartowicz, Monika Staś-Bobis, Błażej Dziuk, Dawid Siodłak

**Affiliations:** 1Faculty of Chemistry and Pharmacy, University of Opole, Oleska 48, 45-052 Opole, Poland; 2Faculty of Chemistry, Wroclaw University of Science and Technology, Wybrzeze Wyspianskiego 27, 50-370 Wroclaw, Poland; 3Faculty of Chemistry, University of Wroclaw, Joliot-Curie 14, 50-383 Wroclaw, Poland

**Keywords:** fungicide, β-methoxyacrylate, non-standard amino acids, dehydroamino acids, dehydroserine, Ramachandran diagram

## Abstract

O-Methyldehydroserine, ΔSer(Me), is a non-standard α,β-dehydroamino acid, which occurs naturally in Cyrmenins with potential pharmaceutical application. The C-terminal part and the side chain of the ΔSer(Me) residue constitute the β-methoxyacrylate unit, responsible for antifungal activity of Cyrmenins. The short model, Ac-ΔSer(Me)-OMe, was analyzed considering the geometrical isomer Z (**1**) and E (**2**). The Ramachandran diagrams were created for both isomers, using quantum chemical calculations, to show possible conformations for isolated molecules (in vacuo), in weakly polar (chloroform) and polar (water) environments. The Ac-(*Z*)-ΔSer(Me)-OMe (**1**) was synthesized and the single-crystal X-ray diffraction analysis together with FT-IR spectra were performed. The detailed analysis of the conformations of the (*Z*)-ΔSer(Me) residue is presented considering the intra- and intermolecular interactions as well as their influence on the β-methoxyacrylate part. It is concluded that the β-methoxyacrylate structural motif is able to maintain a planar geometry, crucial for biological activity, regardless of the conformation adopted by O-methyldehydroserine.

## 1. Introduction

Cyrmenins are naturally occurring peptides produced by *Cystobacter armeniaca* and *Archangium gephyra* (myxobacteria) [1,2]. Structurally, Cyrmenins are N-acyldehydrodipeptide methyl esters, R-CO-ΔAla-(*Z*)-ΔSer(Me)-OMe, containing two non-standard amino acid residues, dehydroalanine and (*Z*)-O-methyldehydroserine, (*Z*)-ΔSer(Me) [3], also like the N-terminal (2*E*,4*Z*)-10-methylundecadienoic, -dodecadienoic, or -11-methyldodecadienoic acid (Figure 1). The structure of Cyrmenins is confirmed by the total synthesis [4,5].

The (*Z*)-ΔSer(Me) residue contains the β-methoxyacrylate structural unit, important for biological activity as a pharmacophore (toxophore). The β-methoxyacrylate was earlier discovered in naturally occurring Strobilurins, Oudemansins, and Myxothiazols (Figure 2) [6]. All these compounds reveal antifungal activity as inhibitors of mitochondrial respiration at the cytochrome bc1 complex [7].

Isolations of Strobilurins and discovery of their antifungal activity resulted in the development of Strobilurin fungicides, known as β-methoxyacrylate fungicides such as Azoxystrobin, Coumoxystrobin, Enoxastrobin, Pyraoxystrobin, Picoxystrobin, or Flufenoxystrobin (Figure 3). They play an important role as antifungal agrochemicals, used as active agents protecting important plants such as cereals (wheat, rice), oilseeds, vegetables (cucumbers, tomatoes), and fruits (grapes) from fungal diseases (downy mildew, blight, powdery mildew, leaf blotch, net blotch, leaf rust, brown rust, and tan spot) (Compendium of Pesticide Common Names) [8].

β-Methoxyacrylate derivatives as cytochrome bc1 inhibitors were recently reviewed [9,10]. The conclusion can be drawn that the β-methoxy group and carbonyl group at the C=C double bond are in transoidal position with respect to each other. In the case of the studied ΔSer(Me) residue, it corresponds to the isomer Z.

Cyrmenins, like other compounds containing a β-methoxyacrylate structural unit, also reveal an antifungal activity (yeasts and filamentous fungi), but no effect on bacteria. Their activity is similar to Strobilurin A. However, they are less toxic for animal cell cultures (mouse cells) [2]. This feature makes them promising precursors of potential antifungal pharmaceuticals.

Cyrmenins belong to peptides. For the peptides, the conformation is important because it determines the spatial arrangement of functional groups, which has a direct impact on their biological activity. The conformation of the whole peptide molecule depends on the conformation adopted by each amino acid residue. Cyrmenins are simple dehydrodipeptides. The conformational properties of dehydroalanine are well known. It has a tendency to adopt the extended conformation C5 (φ, ψ~−180°, 180°) stabilized by the internal N-H⋯O and Cβ-H⋯O hydrogen bonds as well as π-electron cross-conjugation expanded from the Cα=Cβ double bond to the adjacent N- and C-terminal peptide groups, making the whole structure flat [11,12]. However, O-methyldehydroserine is a unique dehydroamino acid, found to date, only in Cyrmenins. No study on its conformational properties has been performed so far. What is important is that the O-methyldehydroserine residue contains the β-methoxyacrylate unit, created by the methoxy group at the carbon atom β and the Cα=Cβ double bond.

The aim of the presented work is to investigate how the conformation adopted by O-methyldehydroserine residue influences the geometry of the β-methoxyacrylate unit, crucial for biological activity of Cyrmenins.

## 2. Results

### 2.1. Theoretical Method

The quantum chemistry calculations are presented first, because they give the overall picture of the conformations, which can possibly be adopted. Then, these results are compared to the experimental X-ray and FT-IR data, which give the conformational preferences of the O-methyldehydroserine residue.

A common approach to assess the conformational properties of selected amino acid residues is the analysis of short-model compounds [13]. The conformational analysis of O-methyldehydroserine residue was performed on two molecular models, Ac-(*Z*)-ΔSer(Me)-OMe (**1**) and Ac-(*E*)-ΔSer(Me)-OMe (**2**) (Figure 4). These models represent the geometrical isomers Z/E resulting from the presence of different substituents and the π-bond between the carbon atoms Cα=Cβ. Assuming that the N-terminal amide and C-terminal ester are typically in trans arrangement, the conformation of the molecule depends on two torsion angles: φ (C-N-Cα-C) and ψ (N-Cα-C-O). The potential energy surfaces E = (φ,ψ) together with the conformations are presented in Figure 5 and Figure 6. The selected parameters, including the values of torsion angles φ and ψ as well as the energy and the population of the conformations (p), are presented in Table 1 and Table 2. The structural parameters for the internal hydrogen bond (X–H⋯A) and carbonyl interactions (C=O ◄⋯► O=C) are presented in Appendix A.

Four conformations were found, regardless of the Z/E isomers: the extended C5, semi-extended β2, helical α, and helical polyproline II-like β. Due to the lack of chirality, each of these conformations have their mirror counterparts with the same energy but opposite sign of the torsion angle values. For clarity of description, the conformations on the left side of the maps are discussed.

For the isomer Z, Ac-(*Z*)-ΔSer(Me)-OMe (**1**), for the isolated molecules, the values of the torsion angles φ and ψ, which determine the geometry of the residue, are as follows: C5 (φ, ψ = −129°, 178°), β2 (φ, ψ = −126°, −2°), α (φ, ψ = −54°, −20°), and β (φ, ψ = −56°, 164°). The simulated increase in the environment polarity (chloroform, water) has rather low impact on the geometry of the conformations. The changes in the torsion angles φ and ψ do not exceed 9° and 5°, respectively. Nevertheless, the values of the torsion angle φ decrease for the extended and semi-extended conformations (C5 and β2) and increase for the helical conformations (α and β). This shows that the N-terminal amide group becomes more perpendicular towards the Cα=Cβ double bond. The values of the torsion angle ψ remain almost unchanged for the conformations C5 and β2. In the case of the helical conformations, the ψ value decreases for the conformation α but increases for the conformation β. This shows that the C-terminal ester group becomes more parallel towards the Cα=Cβ double bond. Consequently, as the polarity of the environment increases, the β-methoxyacrylate unit becomes more flat.

The conformation C5 has the lowest energy for the isolated molecule. However, the energy gap to the highest-in-energy conformation β is relatively small and does not exceed 2 kcal/mol. As a result, the population has a relatively low energy value (estimated at 61%). An increase in the polarity of the environment causes a further decrease in the energy difference between the conformations. In the chloroform environment, the gap in energy does not exceed 0.3 kcal/mol, and ensures that all conformations are accessible and populated. Also, for the water environment, the energy difference (0.5 kcal/mol) is small (Table 1).

For the isomer E, Ac-(*E*)-ΔSer(Me)-OMe (**2**), there are also four conformations, but for the isolated molecule, only the extended and semi-extended conformations, C5 (φ, ψ = −180°, 180°) and β2 (φ, ψ = −180°, 0°), are predicted. In the more polar environment, the helical conformations α (φ, ψ = −66°, −9°) and β (φ, ψ = −66°, 173°) appear. It should be noticed that the geometry of all conformations seems to be resistant to the increase in the polarity of the environment. For the extended conformations C5 and β2, the studied (*E*)-ΔSer(Me) residue is completely flat. For the helical conformations α and β, the N-terminal part tends to be perpendicular in relation to the Cα=Cβ double bond. Again, the C-terminal part containing the β-methoxyacrylate unit is completely or nearly flat. The conformation C5 is the lowest in energy for the isolated molecule and almost completely populated (95%). Nevertheless, the energy gap to the conformation β2 does not exceed 2 kcal/mol. In the low-polar environment, mimicked by the chloroform solvent, the gap in energy decreases to 1.3 kcal/mol, but the conformation C5 remains the first choice (57%). In the more polar water environment, the energy difference further decreases to 0.4 kcal/mol and the conformations with the torsion angle ψ = 180° and 173°, C5 and β, are favored (Table 2).

### 2.2. Crystal Structure Analysis

A single crystal of Ac-(*Z*)-ΔSer(Me)-OMe (**1**) was obtained from a diethyl ether solution. The experimental details of X-ray data collection together with the selected geometric parameters are presented in Appendix A. The molecular structure with the atom numbering scheme is presented in Figure 7. The X-ray diffraction analysis performed at 100 K gave indications on the dihedral angles adopted.

The conformations β (−68.99°, 174.17°) and −β (68.99°, −174.17°) were found in the crystal. The presence of the mirror conformations is typical for dehydroamino acids, which does not have asymmetrical carbon atom alpha. These results confirm the results of quantum chemical calculations performed for the polar environment (water), for which the conformation β is amongst the lowest in energy and equally populated with the conformation C5. The geometry of conformations is quite similar and the difference in values of torsion angles φ and ψ does not exceed 7°.

The β-methoxyacrylate structural unit is almost flat, which can be determined by co-planarity of the Cα=Cβ π-bond and the C-terminal ester carbonyl C=O group, with the absolute value of the torsion angle C3-C2-C1-O1 being equal to 178.98°.

The molecular interactions of the Ac-(*Z*)-ΔSer(Me)-OMe (**1**) molecules (Figure 8) are determined by the N-H…O hydrogen bonds. A linear arrangement can be observed, in which the molecules adopt the conformation β and −β, alternately. Only the N-terminal amide group is involved in the N-H…O hydrogen bonds. The β-methoxyacrylate structural unit creates only weak C-H…O short contacts.

### 2.3. Infrared Absorption Spectroscopy

The conformational properties of the (*Z*)-ΔSer(Me) residue in non-polar or weakly polar solutions were estimated using the model compound Ac-(*Z*)-ΔSer(Me)-OMe (**1**). The spectra in CCl_4_ and CHCl_3_ are presented in Appendix A, respectively. The ν_s_(N-H) stretching mode region was analyzed. The studied molecule has one N-H bond (N-terminal amide group). Therefore, the number of the bands in this region provides information about the number of conformations in the solution. The intensity of the bands gives information about the population of each conformation. The position of the band indicates if the N-H group is involved in the N-H…O hydrogen bond and enables us to estimate the strength of this hydrogen bond. This information allows us to assign the bands to the conformations.

As can be seen in Figure 9, in the non-polar CCl_4_ solution, the ν_s_(N-H) stretching mode region is occupied by two maxima. After deconvolution, the maximum of the band of higher intensity was determined as 3430 cm^−1^, whereas another band was determined as 3444 cm^−1^. Amongst the accessible conformations, the conformation C5 has the strongest N-H…O bond and it is predicted as the lowest in energy. Therefore, the conformation C5 can be assigned to the band at 3430 cm^−1^. Comparison to the theoretical frequencies (Appendix A) shows that the band at 3444 cm^−1^ can be assigned to the conformation β2.

The spectra recorded in the CHCl_3_ solution shows in the ν_s_(N-H) stretching mode region the single band, but expanded towards higher wavenumber values. The deconvolution procedure fits the best when two components are applied with maxima at 3427 and 3433 cm^−1^. Again, the intensity and the position of the band at 3427 cm^−1^ allow us to assign this band to the conformation C5. The band at 3433 cm^−1^, after comparison to the theoretical frequencies, fits to the conformation α. In summary, the FTIR analysis corroborates quantum chemical calculations; C5, the lowest in energy conformation, prevails in non-polar and weakly polar environments.

## 3. Discussion

The existence of the conformations of the ΔSer(Me) residue as well as their relative energies, and thus order of population, can be explained by the presence of internal interactions.

For the isomer Z (**1**), as can be seen in Figure 5, the extended and semi-extended conformations C5 and β2 are stabilized by the N-H…O hydrogen bonds, where the N-terminal amide group is the donor and the C-terminal ester group is the acceptor. The type of the oxygen atom, from the carbonyl or methoxy group, is important because, as the hydrogen bond acceptor, it influences the stability of the hydrogen bond. The oxygen atom from the carbonyl group is a better acceptor than that from the methoxy group, thus creating a stronger hydrogen bond. In results, the conformation C5 has lower energy, and thus, it is more populated than the conformation β2. However, the N-H…O hydrogen bond is also found in the conformation α, where the oxygen atom of the methoxy group of the side chain is involved. Moreover, the conformation α is stabilized by carbonyl interactions, motif II [15]. This explains the second position in the energy order for the isolated molecule. The fourth conformation, β, is stabilized by carbonyl interactions, motif II. Finally, in all four conformations, the Cβ-H…O hydrogen bonds should be considered, where the acceptors are the oxygen atoms of the C-terminal ester bonds. The existence of the stabilizing hydrogen bond and/or carbonyl interactions in all conformations explains their relatively small energy differences.

The increase in the polarity of the environment influences the geometry of the conformations. The N-terminal amide group becomes more perpendicular towards the Cα=Cβ. This can be explained by opening the C=O and N-H of amide towards intermolecular interactions. At the same time, the C-terminal ester group becomes more co-planar with the Cα=Cβ bond. This indicates that this part of the ΔSer(Me) residue does not create any stronger intermolecular interactions, so that it gains stabilization from the π-electron conjugation of the β-methoxy group, Cα=Cβ double bond, and C-terminal ester (Figure 10).

For the isomer E (**2**), there is a quite similar pattern. The conformations C5 and β2 are stabilized by the N-H…O hydrogen bonds, where the N-terminal amide group is the donor and the carbonyl or methoxy oxygen atoms from the C-terminal ester group are acceptors, respectively. Also, the Cβ-H…O hydrogen bonds are present. In contrast, the conformations α and β are stabilized by carbonyl interactions (Figure 6). Nevertheless, the comparison of the energies in hartrees indicates that the transoidal β-methoxyacrylate unit (isomer Z) is predicted as the most preferable (Table 2).

The results obtained from experimental methods corroborate the quantum chemical calculations. In the case of the FT-IR method (Figure 9), the position of the band at 3430–3427 cm^−1^ assigned to the conformation C5 indicates that the strength of the N-H…O hydrogen bond is moderate. The β-methoxy group in the position Z of the side chain imposes a steric hindrance on the N-terminal amide group (torsion angle φ~−125°). Consequently, the N-terminal amide group and C-terminal ester bond are not in a plane, which increases a distance between N-H and C=O and worsens the geometry of the hydrogen bond. The position of the band at 3444 cm^−1^ assigned to the conformation β2 shows that the strength of the N-H…O hydrogen bond can be perceived as very weak. This can be explained by the geometry of the conformation β2, i.e., by the steric hindrance of the β-methoxy group, as in the case of the conformation C5. Also, the donor is the oxygen atom from the C-terminal C-O methoxy group. However, the band at 3433 cm^−1^ assigned to the conformation α in the chloroform solution indicates that the N-H…O hydrogen bond between the N-terminal N-H group and C-O bond of the β-methoxy group in the position Z is stronger. The crystal structure analysis of Ac-(*Z*)-ΔSer(Me)-OMe (**1**) shows that only the N-terminal amide group is involved in the N-H…O hydrogen bonds and the torsion angle φ has the highest value (−68.99°) as compared to that predicted for the chloroform and water environments. The consequence of the relatively weak intramolecular interactions is the small energy difference between the conformations. The presence of the conformation β in the crystal state can be explained by the fact that it forms stronger intermolecular N-H…O hydrogen bonds. Thus, all possible conformations of the ΔSer(Me) residue are available.

The β-methoxyacrylate structural motif included in the O-methyldehydroserine residue has a tendency to adopt a planar geometry, regardless of the conformation adopted. The analysis of the value of the torsion angle ψ (O1-C1-C2-N1, Figure 7) shows that it does not deviate more than 20° from the planar structure (180° for the conformations C5 and β as well as 0° for the conformation β2 and α) (Table 1). Also, a comparison to the structurally closest analog, dehydrobutyrine residue, with C-terminal ester [16] or amide [17], shows that the C-terminus is more flat for the studied ΔSer(Me) residue, especially for the isomer Z. This can be caused by the oxygen atom in the position β of the side chain, which is able to participate in the π-electron cross-conjugation with the Cα=Cβ double bond and C-terminal ester group, as well as a smaller steric hindrance compared to the β-methyl group of the dehydrobutyrine residue.

The World Protein Data Bank [18] shows the structures of proteins containing as ligands β-methoxyacrylate fungicide, like Azoxystrobin [19,20]. As it is presented in Figure 11, the β-methoxyacrylate unit is planar and the β-methoxy group is in the position E in relation to the ester group. This corresponds to the isomer Z for the ΔSer(Me) residue. It indicates that both planarity of the β-methoxyacrylate unit and the position of the β-methoxy group are crucial for antifungal activity. Also, the neighboring benzene ring does not participate in a larger delocalized electron system, but instead, tends to adopt a perpendicular position. The same trends can be seen in the conformations of the ΔSer(Me) residue in the polar environment. As can be seen, the rotation of the ester group is possible, which indicates a conformational flexibility.

## 4. Materials and Methods

### 4.1. Quantum Chemical Calculations

The model compounds, Ac-(*Z*)-ΔSer(Me)-OMe (**1**) and Ac-(*E*)-ΔSer(Me)-OMe (**2**), were calculated by the DFT method using the Gaussian 16 package [21]. The initial structures were prepared with GaussView5 [22]. The configuration trans (ω0 ≈ 180°) of amide and ester groups was set. The potential energy surfaces E = f(φ,ψ) were obtained at the M06-2X/6-31+G(d,p) level of theory [23], in the gas phase, and then in chloroform and water (constrained optimization). The values of the φ and ψ dihedral angles were changed in steps of 30°. Because of the achirality of dehydroamino acids, each structure has a mirror counterpart with the same energy, but opposite torsion angles (φ, ψ = −φ, −ψ). This reduces the number of grid point structures necessary to create the maps to 91 each. The solvent effect was simulated with the self-consistent reaction field (SCRF) using the conductor-like polarizable continuum model (CPCM). All potential energy minima localized in the maps were fully optimized by using a larger basis set, 6-311+G(d,p). Unconstrained optimizations were followed by a vibrational analysis to ensure that the resulting structures are true energy minima and to obtain the zero-point vibrational energies (ZPVEs) and Gibbs energies (298.15 K, 1.0 atm). The population of the conformations (p) was calculated at 300 K, where RT = 0.595 kcal/mol according to the following equations: *p*rel = 100% exp(−ΔG/*R**T*), *p* = *p*rel/∑conformers*p*rel × 100% [24,25]. The names of the conformations are based on the Scarsdale nomenclature [26].

### 4.2. Synthesis

Ac-(*Z*)-ΔSer(Me)-OMe (**1**) was synthesized from glycine in a four-step reaction (Appendix A) according to the procedure described in a patent (WO 2012/041986). The detailed synthetic procedure is presented in the Appendix A. The ^1^H and ^13^C NMR spectra of the compounds are presented in Appendix A. The NOE difference method was applied using the standard programs to determine the isomer Z. A photoisomerization reaction was performed using Ac-(*Z*)-ΔSer(Me)-OMe (**1**) as the substrate in order to obtain the isomer E, according to a method elaborated on previously [27]. Unfortunately, the procedure led to the decomposition of the substrate to ammonium oxalate.

### 4.3. NMR Spectra

A Bruker Ultrashield 400 (Bruker 2005, Berlin, Germany) spectrometer was used. Bruker (TopSpin Version 1.3) software was also used, operating at 400 MHz (^1^H) and 101 MHz (^13^C). The spectra were recorded in DMSO-d6 (internal TMS standard) at room temperature.

### 4.4. FTIR Spectra

A Nicolet Nexus 2002 FT-IR spectrometer (New Castle, DE, USA) was used. Measurement conditions involved a dry nitrogen atmosphere and KBr liquid cell (0.01 mm), and for each measurement, there were 20 scans with a 2 cm^−1^ resolution (spectral resolution: 0.482 cm^−1^) in the spectral range 400–4000 cm^−1^. The Ac-(*Z*)-ΔSer(Me)-OMe (**1**) solutions in CCl_4_ and CHCl_3_ (distilled over P_2_O_5_) were in three different concentrations: 0.5, 1.0, and 2.0 mg/mL each. The spectra processing and peak deconvolution were conducted using Fityk software (version 1.3.1) [28], applying the Voigt function. 

### 4.5. Crystal Structure Analysis

Data from the X-ray diffraction of the single crystal for Ac-(*Z*)-ΔSer(Me)-OMe (**1**) were collected on a Rigaku Oxford Diffraction XtaLAB SynergyR DW diffractometer (Tokyo, Japan) equipped with a HyPix ARC 150° Hybrid Photon Counting (HPC) detector using MoKα (λ = 0.71073 Å) at 100 K. The corrections to the Lorentz and polarization factors were applied to the reflection intensities (CrysAlis CCD; Oxford Diffraction Ltd.: Abingdon, UK, (2002). CrysAlis RED; Oxford Diffraction Ltd.: Abingdon, UK, 2002). Data were processed using CrysAlisPro software (version 1.171.39.44, CrysAlis PRO; Rigaku Oxford Diffraction. Rigaku Oxford Diffraction Ltd., Yarnton, UK, 2015, 2022). The structures were solved by direct methods using SHELXS and refined by full-matrix least-squares methods based on F2 using SHELXL [29,30]. All non-hydrogen atoms were located from difference Fourier synthesis and refined by the least-squares method in the full-matrix anisotropic approximation. The crystallographic data for compounds and details of the X-ray experiment are collected in the Appendix A. The structure drawings in ESI were prepared by using the Mercury program [31]. The coordinates of atoms and other parameters for structures were deposited with the Cambridge Crystallographic Data Centre: 2394235 for (**1**)—12 Union Road, Cambridge CB2 1EZ, UK (Fax,_44-(1223)336-033, E-mail: deposit@ccdc.cam.ac.uk).

## 5. Conclusions

The O-methyldehydroserine residue, ΔSer(Me), is able to adopt four conformations: the extended C5, semi-extended β2, helical α, and helical polyproline II-like conformation β. The lack of chirality of the carbon α ensures that the mirror counterparts of these conformations are equally accessible (-C5, -β2, -α, and -β) as it is presented in Ramachandran diagrams. The β-methoxy group in the position Z or E influences the value of the torsion angles φ and ψ of the main chain. The quantum chemical calculations for weakly polar (chloroform) and polar (water) environments show the following φ and ψ torsion angle values: for the isomer Z—C5 (φ, ψ = −121°, 180°), β2 (φ, ψ = −119°, 0°), α (φ, ψ = −59°, −16°), and β (φ, ψ = −61°, 168°), and for the isomer E—C5 (φ, ψ = −180°, 180°) and β2 (φ, ψ = −180°, 0°), and α (φ, ψ = −67°, −9°) and β (φ, ψ = −68°, 173°). The X-ray diffraction analysis performed at 100 K for a single crystal of Ac-(*Z*)-ΔSer(Me)-OMe (**1**) shows the conformations β (−68.99°, 174.17°) and −β (68.99°, −174.17°). The infrared absorption spectra for Ac-(*Z*)-ΔSer(Me)-OMe (**1**) show the conformations C5 and β2 in the non-polar CCl_4_ solution as well as the conformations C5 and α in the weakly polar CHCl_3_ solution. Thus, the results obtained using experimental methods corroborate those predicted by the quantum chemical calculations. It is also shown that all conformations should be accessible.

In summary, the (*Z*)-ΔSer(Me) residue has a tendency to adopt in a weakly polar environment primarily the conformation C5, followed by the conformation β2, which are mainly stabilized by intramolecular interactions. As polarity of the environment increases, the conformation α and then β should prevail as they will be stabilized by intermolecular interactions involving mainly the N-terminal amide group. The β-methoxyacrylate structural motif should maintain a planar geometry, not involved in any stronger intramolecular or intermolecular interactions. Thus, the conformation properties of the ΔSer(Me) residue are flexible and should not affect the biological function of the β-methoxyacrylate toxophore.

## Figures and Tables

**Figure 1 ijms-26-00340-f001:**
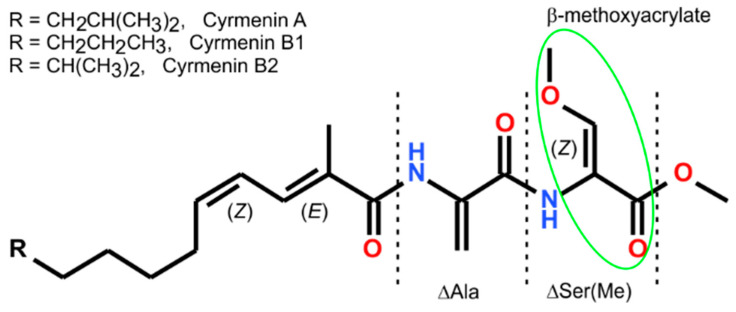
Structure of Cyrmenins, antifungal peptides, with marked β-methoxyacrylate unit.

**Figure 2 ijms-26-00340-f002:**
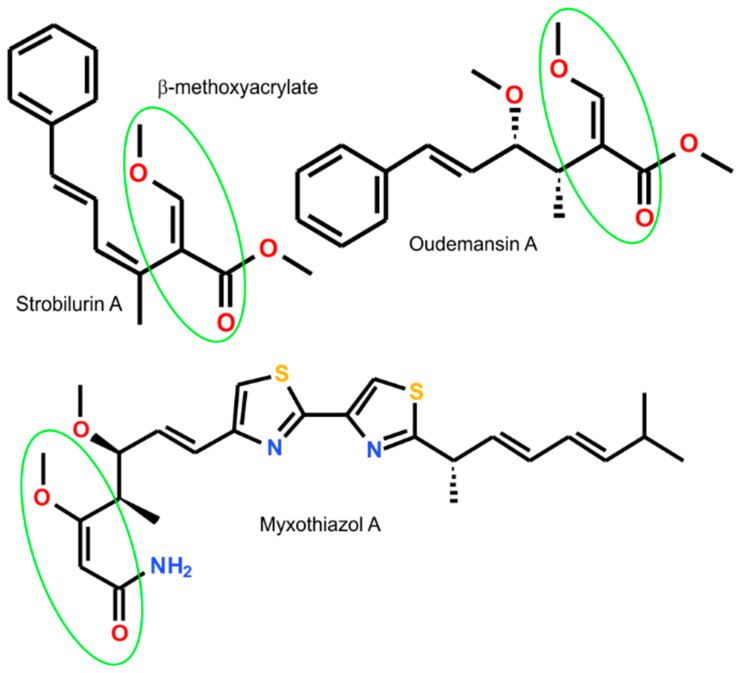
The structure of Strobilurin A, Oudemansin A, and Myxothiazol A, naturally occurring precursors of β-methoxyacrylate fungicides.

**Figure 3 ijms-26-00340-f003:**
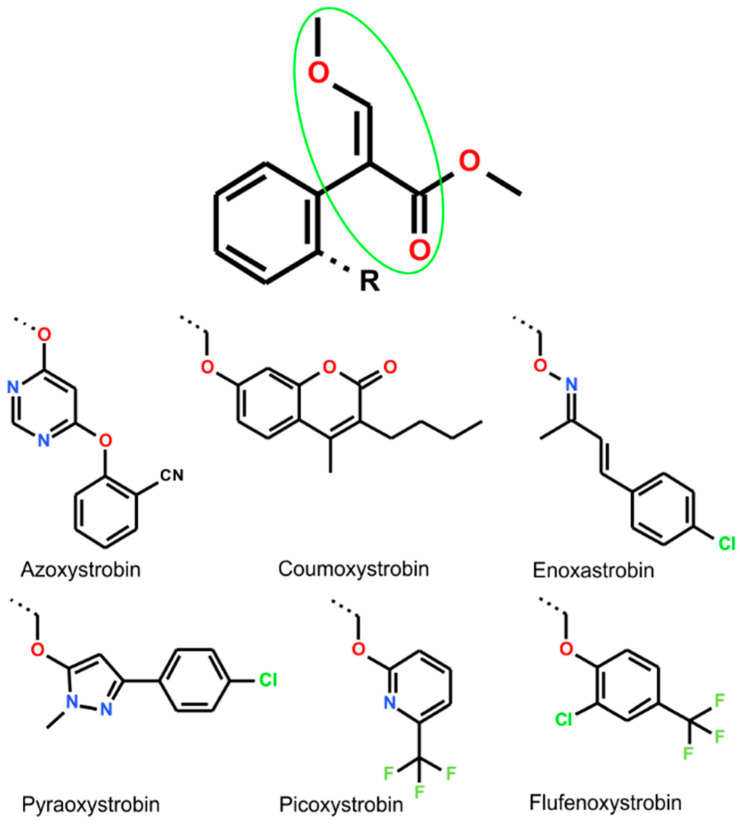
The structure of Azoxystrobin, Coumoxystrobin, Enoxastrobin, Pyraoxystrobin, Picoxystrobin, and Flufenoxystrobin, designed β-methoxyacrylate fungicides commonly applied as agrochemicals.

**Figure 4 ijms-26-00340-f004:**
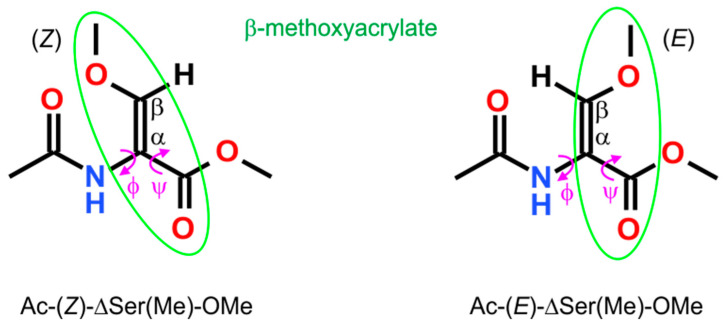
Two molecular models, Ac-(*Z*)-ΔSer(Me)-OMe (**1**) and Ac-(*E*)-ΔSer(Me)-OMe (**2**), applied in the conformational analysis of O-methyldehydroserine residue. The geometrical isomers Z/E result from the presence of different substituents and the π-bond between the carbon atoms Cα=Cβ. Two main torsion angles φ and ψ are shown together with the β-methoxyacrylate unit.

**Figure 5 ijms-26-00340-f005:**
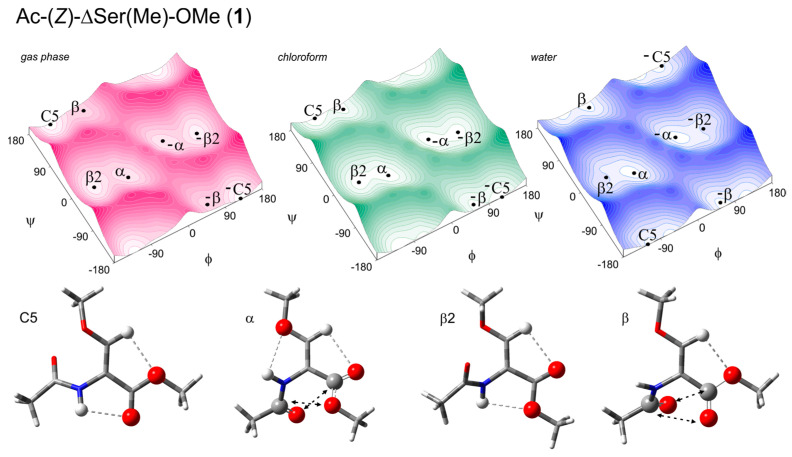
The potential energy surfaces E = (φ,ψ) of Ac-(*Z*)-ΔSer(Me)-OMe (**1**) calculated via the M06-2X/6-31+G(d.p) method in the gas phase, chloroform, and water environments. Energy contours are plotted every 1 kcal/mol. Below the maps are the conformations optimized at the M06-2X/6-311+G(d,p) level of theory with the hydrogen bonds (⋯) [14] and electrostatic interactions (◄⋯►) [15] created within the residue.

**Figure 6 ijms-26-00340-f006:**
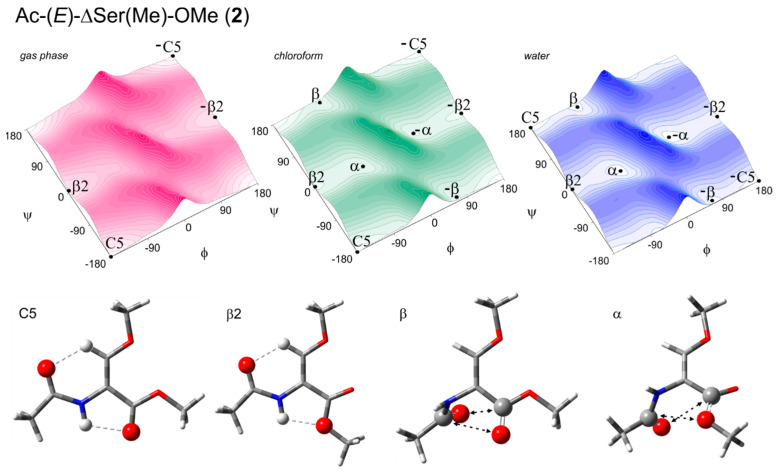
The potential energy surfaces E = (φ,ψ) of Ac-(*E*)-ΔSer(Me)-OMe (**2**) calculated via the M06-2X/6-31+G(d.p) method in the gas phase, chloroform, and water environments. Energy contours are plotted every 1 kcal/mol. Below the maps are the conformations optimized at the M06-2X/6-311+G(d,p) level of theory with the hydrogen bonds (⋯) [14] and electrostatic interactions (◄⋯►) [15] created within the residue.

**Figure 7 ijms-26-00340-f007:**
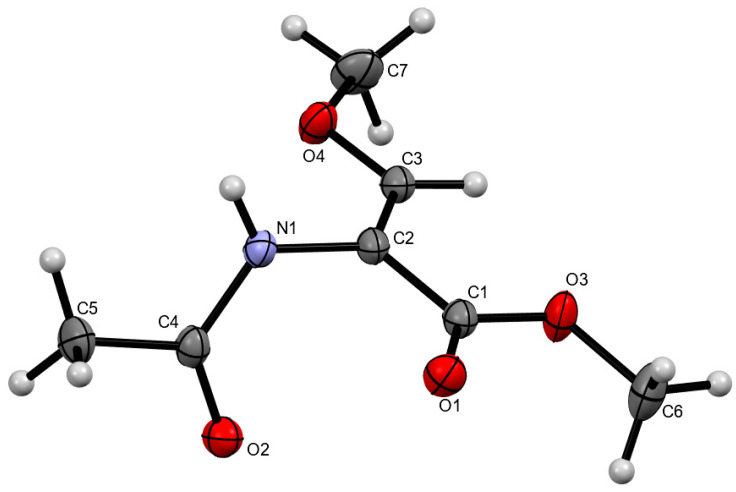
The molecular structure of Ac-(*Z*)-ΔSer(Me)-OMe (**1**) in the asymmetric part of the unit cell. Displacement ellipsoids are drawn at the 50% probability level.

**Figure 8 ijms-26-00340-f008:**
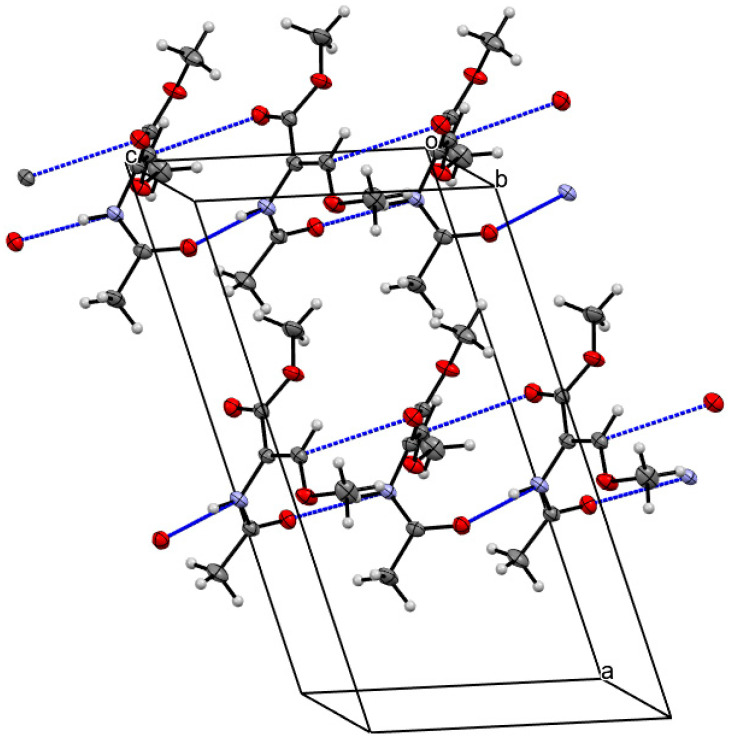
Molecular interactions of the Ac-(*Z*)-ΔSer(Me)-OMe (**1**) molecules. The hydrogen bonds are visualized by dotted lines. Mercury 3.8 (Build RC2) (http://www.ccdc.cam.ac.uk/mercury/, accessed on 29 October 2024) was applied. The hydrogen bonds rely on the sum of van der Waals radii of the atom involved.

**Figure 9 ijms-26-00340-f009:**
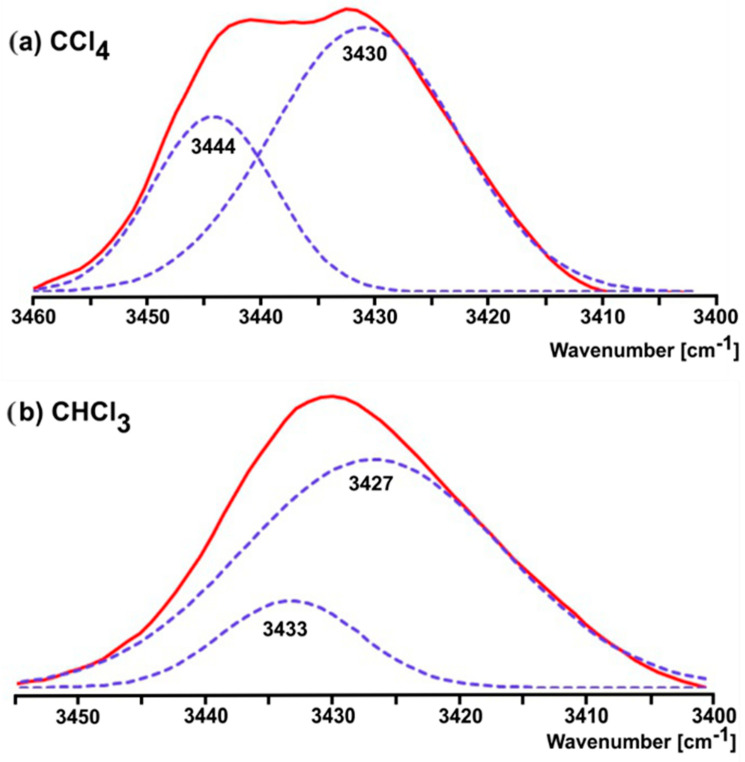
FTIR spectra for Ac-(*Z*)-ΔSer(Me)-OMe (**1**) in region ν_s_(N-H): (**a**) CCl_4_, (**b**) CHCl_3_. The dashed lines, obtained by the curve-fitting procedure, present bands for each conformation.

**Figure 10 ijms-26-00340-f010:**
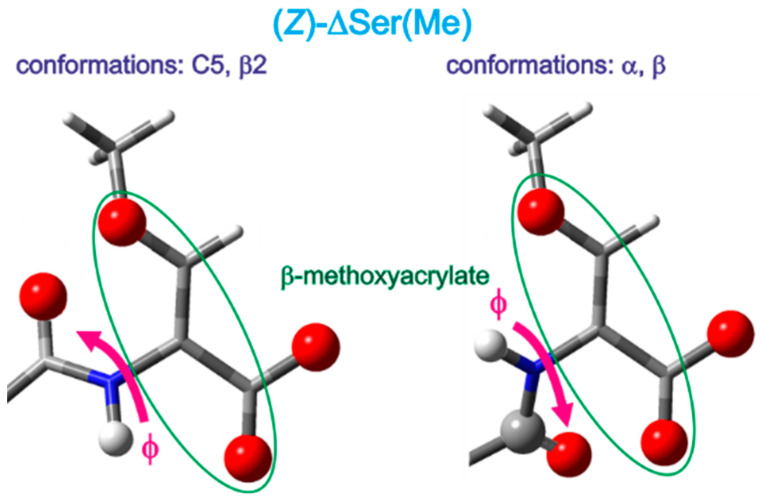
A schematic representation of the conformational changes in the ΔSer(Me) residue with increasing environmental polarity.

**Figure 11 ijms-26-00340-f011:**
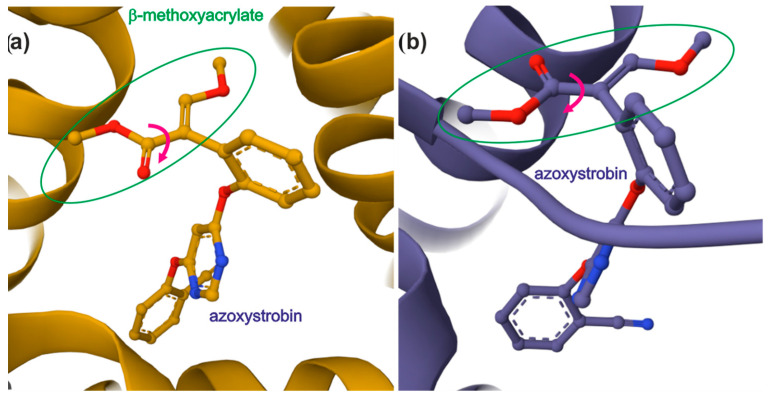
Cytochrome bc1 complex with Azoxystrobin: (**a**) ref. [18], (**b**) ref. [19]. β-methoxyacrylate unit circled with green oval. Possible rotation of ester group marked with pink arrow. Structures were retrieved using 3D Viewers tool accessible at PDB website (https://www.rcsb.org, accessed on 30 December 2024).

**Table 1 ijms-26-00340-t001:** Selected Parameters for Conformations of Ac-(*Z*)-ΔSer(Me)-OMe (**1**).

Ac-(*Z*)-ΔSer(Me)-OMe (1)					
Conformation	φ [°]	ψ [°]	G [Hartrees]	ΔG [kcal]	p [%]
gas phase					
C5	−129.0	178.1	−628.79563	0.00	61.42
α	−53.6	−20.0	−628.79452	0.70	19.05
β2	−126.4	−2.4	−628.79439	0.78	16.57
β	−56.3	163.9	−628.79276	1.81	2.96
chloroform					
α	−57.9	−16.6	−628.80850	0.00	28.60
C5	−122.2	179.8	−628.80844	0.04	26.63
β	−60.7	167.2	−628.80843	0.04	26.52
β2	−119.7	−0.4	−628.80808	0.27	18.25
water					
C5	−120.2	−179.3	−628.81339	0.00	33.62
β	−62.1	168.4	−628.81337	0.01	32.82
α	−59.2	−15.7	−628.81289	0.31	19.82
β2	−117.7	0.2	−628.81254	0.53	13.74

Each calculated conformation has its mirror counterpart. Optimized via the M06-2X/6-311+G(d,p) method (SCRF, CPCM). The population of the conformations (p).

**Table 2 ijms-26-00340-t002:** Selected Parameters for Conformations of Ac-(*E*)-ΔSer(Me)-OMe (**2**).

Ac-(*E*)-ΔSer(Me)-OMe (2)					
Conformation	φ [°]	ψ [°]	G [Hartrees]	ΔG [kcal]	p [%]
gas phase					
C5	−180.0	−180.0	−628.79370	0.00	94.86
β2	−180.0	0.0	−628.79093	1.74	5.14
chloroform					
C5	−180.0	−180.0	−628.80438	0.00	57.02
β2	−180.0	0.0	−628.80334	0.65	19.12
β	−68.4	173.3	−628.80323	0.72	16.96
α	−66.8	−8.2	−628.80237	1.26	6.90
water					
β	−67.7	172.8	−628.80794	0.00	32.17
C5	−180.0	180.0	−628.80773	0.13	25.83
α	−66.0	−10.1	−628.80765	0.18	23.58
β2	−180.0	0.0	−628.80741	0.33	18.42

Each calculated conformation has its mirror counterpart. Optimized via the M06-2X/6-311+G(d,p) method (SCRF, CPCM). The population of the conformations (p).

## Data Availability

The original contributions presented in this study are included in the article/Appendix A. Further inquiries can be directed to the corresponding author(s).

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
