# Peer review of "Insight into the Structure of Antifungal Cyrmenins: Conformational Studies of Unique Dehydroamino Acid, O-Methyldehydroserine"

_ijms, 2025, doi:10.3390/ijms26010340_

Round 1
Reviewer 1 Report
Comments and Suggestions for Authors
This article describes a structural understanding of the partial structure responsible for bioactivity of Cyrmenins, which are natural peptides that exhibit antifungal activity. Supposing that the serine derivatives that share the important structural feature may represent the bioactivity of the whole peptide, they prepared two model compounds. Their comprehensive structural analysis using NMR, IR, and X-ray was discussed with the help of quantum mechanics calculations, succeeding in proof that the flatness of the structure enables complexation for Cyrmenins. Therefore, the reviewer would recommend publication in the journal, if the authors update the following minor points.
(1) In line 91 of page 3, complete structures for compounds (1) and (2) should be shown. There are only structures of relevant compounds in the present manuscript, which forces readers try to draw concrete structures themselves when following the discussion.
(2) In line 95 of page 3, please define the angles phi and psi exactly in the manuscript. Otherwise, it is hard to follow discussion.
(3) In line 320 of page 10, the important word "NMR" is missing after 1H and 13C.
(4) In line 366 of page 11, "in for" should be "for".
(5) In line 375 of page 12, "mainly N-terminal" should be "mainly in N-terminal.
Author Response
We would like to thank you for taking the time to evaluate our submitted manuscript.
The following responses are provided at the reviewer's request.
(1) In line 91 of page 3, complete structures for compounds (1) and (2) should be shown. There are only structures of relevant compounds in the present manuscript, which forces readers try to draw concrete structures themselves when following the discussion.
Response. Thank you for this valuable remark. The new figure (Figure 4) was introduced in the text. This figure contains the structures of the studied models, Ac-(Z)-ΔSer(Me)-OMe (1) and Ac-(E)-ΔSer(Me)-OMe (2). In addition, the torsion angles φ and ψ, the atoms α and β, and the β-metoxyacrylate unit are shown.
(2) In line 95 of page 3, please define the angles phi and psi exactly in the manuscript. Otherwise, it is hard to follow discussion.
Response. Thank you for this remark. The explanation of the torsion angles phi and psi is given. In addition, the graphical visualization is given in the new figure (Figure 4).
(3) In line 320 of page 10, the important word "NMR" is missing after 1H and 13C.
Response. NMR word was introduced. Thank you for this minor, but still important remark.
(4) In line 366 of page 11, "in for" should be "for".
Response. The sentence was corrected. Thank you for this minor, but still important remark.
(5) In line 375 of page 12, "mainly N-terminal" should be "mainly in N-terminal.
Response. The sentence was corrected. Thank you for this minor, but still important remark.
Reviewer 2 Report
Comments and Suggestions for Authors
The article “Insight Into the Structure of Antifungal Cyrmenins. Conformational Studies of Unique Dehydroamino Acid, O-methyldehydroserine” is devoted to theoretical studying a Z- and E- isomers of non-standard α,β-dehydroamino acid the O-methyldehydroserine. This study is undoubtedly novel since there are not many studies on the amino acid O-methyldehydroserine.
Minor comments:
1. The article has grammatical errors, so revising English grammar is necessary.
2. The section “3. Discussion”: specify "Protein Database" in more detail. In this case, it is the World Protein Data Bank (wwPDB). This World Protein Data Bank needs citation:
[Berman, H., Henrick, K. & Nakamura, H. Announcing the worldwide Protein Data Bank. Nat Struct Mol Biol 10, 980 (2003). https://doi.org/10.1038/nsb1203-980]
3. What program was used to obtain Fig. 10? Indicate this program (name, manufacturer, citation).
4. To state "....the β-methoxyacrylate unit is planar and the β-methoxy group is in the position E in relation to the ester group" it is necessary to analyze and present several protein structures containing azoxystrobin. For example: RCSB PDB - PDB ID: 1SQB, 6NHG, etc.
5. In section "4.2. Synthesis" it would be appropriate to place the scheme of synthesis of Ac-(Z)-ΔSer(Me)-OMe (1); to describe in more detail the procedure for obtaining its isomer E, and to add its spectral characteristics.
6. In SM, in the section "Photoisomerization" it is stated:
"Crystallographic analysis showed that the obtained product was not Ac-(E)-ΔSer(OMe)-OMe, as expected, but ammonium oxalate."
Please explain in more detail. It turns out that Ac-(E)-ΔSer(OMe)-OMe does not exist (or is degrades quickly)?
7. The article's title, "...Structure of Antifungal Cyrmenins...." may suggests the need for a fragment of experimental (antifungal) studies to confirm the theory.
The work was done thoroughly and left a positive impression, but several positions require clarification.

The article “Insight Into the Structure of Antifungal Cyrmenins. Conformational Studies of Unique Dehydroamino Acid, O-methyldehydroserine” is devoted to theoretical studying a Z- and E- isomers of non-standard α,β-dehydroamino acid the O-methyldehydroserine. This study is undoubtedly novel since there are not many studies on the amino acid O-methyldehydroserine.
Minor comments:
1. The article has grammatical errors, so revising English grammar is necessary.
2. The section “3. Discussion”: specify "Protein Database" in more detail. In this case, it is the World Protein Data Bank (wwPDB). This World Protein Data Bank needs citation:
[Berman, H., Henrick, K. & Nakamura, H. Announcing the worldwide Protein Data Bank. Nat Struct Mol Biol 10, 980 (2003). https://doi.org/10.1038/nsb1203-980]
3. What program was used to obtain Fig. 10? Indicate this program (name, manufacturer, citation).
4. To state "....the β-methoxyacrylate unit is planar and the β-methoxy group is in the position E in relation to the ester group" it is necessary to analyze and present several protein structures containing azoxystrobin. For example: RCSB PDB - PDB ID: 1SQB, 6NHG, etc.
5. In section "4.2. Synthesis" it would be appropriate to place the scheme of synthesis of Ac-(Z)-ΔSer(Me)-OMe (1); to describe in more detail the procedure for obtaining its isomer E, and to add its spectral characteristics.
6. In SM, in the section "Photoisomerization" it is stated:
"Crystallographic analysis showed that the obtained product was not Ac-(E)-ΔSer(OMe)-OMe, as expected, but ammonium oxalate."
Please explain in more detail. It turns out that Ac-(E)-ΔSer(OMe)-OMe does not exist (or is degrades quickly)?
7. The article's title, "...Structure of Antifungal Cyrmenins...." may suggests the need for a fragment of experimental (antifungal) studies to confirm the theory.
The work was done thoroughly and left a positive impression, but several positions require clarification.
Author Response
We would like to thank you for taking the time to evaluate our submitted manuscript.
The following responses are provided at the reviewer's request.
- The article has grammatical errors, so revising English grammar is necessary.
Response. Thank you for this remark. The whole manuscript was checked thoroughly after introduction of changes on order to revise English grammar.
- The section “3. Discussion”: specify "Protein Database" in more detail. In this case, it is
the World Protein Data Bank (wwPDB). This World Protein Data Bank needs citation:
[Berman, H., Henrick, K. & Nakamura, H. Announcing the worldwide Protein Data Bank.
Nat Struct Mol Biol 10, 980 (2003). https://doi.org/10.1038/nsb1203-980]
Response. Thank you for this important remark. The proper name of this database as well as citation was introduced in the text.
- What program was used to obtain Fig. 10? Indicate this program (name, manufacturer,
citation).
Response. Thank you for this remark. The figures was created using 3D Viewers tool accessible at the PDB website (https://www.rcsb.org). This information was added in the legend of Figure 11.
- To state "....the β-methoxyacrylate unit is planar and the β-methoxy group is in the
position E in relation to the ester group" it is necessary to analyze and present several
protein structures containing azoxystrobin. For example: RCSB PDB - PDB ID: 1SQB,
6NHG, etc.
Response. Thank you for this valuable remark. A new structure was added to Figure 11. This structure was chosen because, not only it shows the planarity of the β-methoxyacrylate unit, but also there is a rotation of ester group, which makes the conformational changes accessible.
- In section "4.2. Synthesis" it would be appropriate to place the scheme of synthesis of
Ac-(Z)-ΔSer(Me)-OMe (1); to describe in more detail the procedure for obtaining its isomer
E, and to add its spectral characteristics.
Response. Thank you for this remark. The scheme of synthesis of Ac-(Z)-ΔSer(Me)-OMe (1) was prepared, however, we decide to place it in Supplementary Materials as Figure 3S.
- In SM, in the section "Photoisomerization" it is stated:
"Crystallographic analysis showed that the obtained product was not Ac-(E)-ΔSer(OMe)-
OMe, as expected, but ammonium oxalate."
Please explain in more detail. It turns out that Ac-(E)-ΔSer(OMe)-OMe does not exist (or is
degrades quickly)?
Response. Thank you for this remark. Despite of our aims, we were not able in this study to obtain Ac-(E)-ΔSer(OMe)-OMe. The procedure applied in previous work, which concerns Ac-ΔAla(βCl)-NHMe, failed. The synthesis of Ac-(Z)-ΔSer(Me)-OMe (1) is also disappointing. At least in our hands, the opening of the oxazolone results in very low yield. It seems that a new method of synthesis must be elaborated, however, it is the subject of a new work.
7.The article's title, "...Structure of Antifungal Cyrmenins...." may suggests the need for a
fragment of experimental (antifungal) studies to confirm the theory.
Our intention was to emphasize that the studied dehydroamino acid residue is part of Cyrmenins, which exhibit antifungal properties. Therefore, we keep the title of the article unchanged. Experimental (antifungal) studies are a valuable proposition, but they should be performed on a series of compounds containing the ΔSer(Me) residue. This cannot be done until the problem of the synthesis of the ΔSer(Me) residue is solved.